# Empagliflozin Protects HK-2 Cells from High Glucose-Mediated Injuries via a Mitochondrial Mechanism

**DOI:** 10.3390/cells8091085

**Published:** 2019-09-14

**Authors:** Wen-Chin Lee, You-Ying Chau, Hwee-Yeong Ng, Chiu-Hua Chen, Pei-Wen Wang, Chia-Wei Liou, Tsu-Kung Lin, Jin-Bor Chen

**Affiliations:** 1Division of Nephrology, Department of Internal Medicine, Kaohsiung Chang Gung Memorial Hospital and Chang Gung University College of Medicine, Kaohsiung 83301, Taiwan; leewenchin@gmail.com (W.-C.L.); stan@cgmh.org.tw (H.-Y.N.); becky0917@cgmh.org.tw (C.-H.C.); 2Centre for Cardiovascular Science, The Queen’s Medical Research Institute, University of Edinburgh, Edinburgh EH16 4TJ, UK; You-Ying.Chau@ed.ac.uk; 3Division of Metabolism, Department of Internal Medicine, Kaohsiung Chang Gung Memorial Hospital and Chang Gung University College of Medicine, Kaohsiung 83301, Taiwan; wangpw@ms18.hinet.net; 4Department of Neurology, Kaohsiung Chang Gung Memorial Hospital and Chang Gung University College of Medicine, Kaohsiung 83301, Taiwan; cwliou@ms22.hinet.net (C.-W.L.); tklin@adm.cgmh.org.tw (T.-K.L.)

**Keywords:** empagliflozin, mitochondria, proximal tubular cell, diabetic kidney disease

## Abstract

Empagliflozin is known to retard the progression of kidney disease in diabetic patients. However, the underlying mechanism is incompletely understood. High glucose induces oxidative stress in renal tubules, eventually leading to mitochondrial damage. Here, we investigated whether empagliflozin exhibits protective functions in renal tubules via a mitochondrial mechanism. We used human proximal tubular cell (PTC) line HK-2 and employed western blotting, terminal deoxynucleotidyl transferase dUTP nick end labelling assay, fluorescence staining, flow cytometry, and enzyme-linked immunosorbent assay to investigate the impact of high glucose and empagliflozin on cellular apoptosis, mitochondrial morphology, and functions including mitochondrial membrane potential (MMP), reactive oxygen species (ROS) production, and adenosine triphosphate (ATP) generation. We found that PTCs were susceptible to high glucose-induced mitochondrial fragmentation, and empagliflozin ameliorated this effect via the regulation of mitochondrial fission (FIS1 and DRP1) and fusion (MFN1 and MFN2) proteins. Empagliflozin reduced the high glucose-induced cellular apoptosis and improved mitochondrial functions by restoring mitochondrial ROS production, MMP, and ATP generation. Our results suggest that empagliflozin may protect renal PTCs from high glucose-mediated injuries through a mitochondrial mechanism. This could be one of the novel mechanisms explaining the benefits demonstrated in EMPA-REG OUTCOME trial.

## 1. Introduction

Diabetic kidney disease (DKD) affects approximately 40% of patients with diabetes and has emerged as a leading cause of chronic kidney disease worldwide [1,2,3]. Empagliflozin, one of the sodium-glucose cotransporter 2 (SGLT2) inhibitors, has recently been reported to retard the progression of DKD in EMPA-REG OUTCOME trial [4]. Animal studies have shown that empagliflozin improves renal functions [5] and is associated with reduced hyperfiltration, albuminuria [6], and ameliorated pathology [7]. However, the mechanisms underlying these effects remain largely unknown. The proposed mechanisms include weight loss, natriuresis, and renal tubuloglomerular feedback [8].

The production of reactive oxygen species (ROS) is increased in diabetes [9], resulting in mitochondrial insult and altered mitochondrial genetics and functions. This mitochondriopathy has emerged as a crucial phenomenon in DKD, owing to known changes in the mitochondrial DNA copy number in the leukocytes [10], altered expression of mitochondrial proteins in DKD patient kidneys, and enlarged mitochondria in the renal proximal tubules of animal models and human renal biopsy tissues [11]. 

Oxidative stress-mediated mitochondrial injuries result in cellular dysfunction and even death via apoptotic pathways. The mitochondrial morphology is shown to be altered in response to these processes. Mitochondrial morphological dynamics are determined by the balance between two opposing processes, namely, fission and fusion. The key proteins involved in mitochondrial fusion include mitofusin 1 (MFN1) and mitofusin 2 (MFN2) [12,13], while dynamin-related protein-1 (DRP1) and mitochondrial fission 1 protein (FIS1) are associated with the mitochondrial fission. High glucose induces mitochondrial fragmentation and cellular apoptosis in renal proximal tubules [14]. As renal proximal tubular cells (PTCs) are the direct site of action for SGLT2 inhibitors, here we used human kidney PTCs to investigate whether SGLT2 inhibitors offer direct renal protection via a mitochondrial mechanism.

## 2. Materials and Methods

### 2.1. Cell Culture and Chemicals

HK-2 cells (ATCC CRL-2190, BCRC 60097, Taiwan) were incubated in culture plates without filters in a humidified atmosphere with 5% CO_2_ at 37 °C. The culture medium comprised of 500 mL of keratinocyte medium (Gibco; Thermo Fisher Scientific, Inc., Waltham, MA, USA), 25 mL fetal bovine serum (FBS; Gibco; Thermo Fisher Scientific, Inc., Waltham, MA, USA), and 5 mL penicillin/streptomycin (Gibco; Thermo Fisher Scientific, Inc., Waltham, MA, USA). All chemicals were purchased from Sigma-Aldrich (St. Louis, MO, USA) unless otherwise stated. The concentrations of empagliflozin (Boehringer Ingelheim, Ingelheim, Germany) used were based on the published report and preclinical studies performed at Boehringer Ingelheim [15,16].

### 2.2. Cell Viability Assay

Crystal violet assay was performed to examine the effect of empagliflozin on cell viability. Cell viability was measured using previously described protocols [14]. Cells were seeded in 96-well plates (5 × 10^3^ cells/well). After an overnight incubation, cells were treated with high (30 mM) or normal (5 mM) concentration of glucose for 24 h. [15]. Cells were treated with 0.5% crystal violet reagent in phosphate-buffered saline (PBS) for 2 h at 37 °C. The cells were washed 10 times with PBS and air-dried, followed by incubation with 100 mL of dimethyl sulfoxide for 1–2 h for complete dissolution of the dye. The absorbance of the reaction product was measured at 570 nm using a microplate reader.

### 2.3. Western Blot Analysis

Cells were seeded at a density of 3 × 10^5^ cells/well in a six-well plate. After an overnight incubation, cells were incubated in normal or high glucose medium. Western blot analysis was carried out as previously described [17]. In brief, cell pellets were homogenized in a buffer and centrifuged at 14,000× *g*. Protein (40 µg) from the supernatant of each sample was separated with sodium dodecyl sulphate polyacrylamide gel electrophoresis (SDS-PAGE) and transferred onto polyvinylidene difluoride membranes by electrophoresis. The membranes were blocked in TBST buffer for 1 h at room temperature. The blots were probed with primary anti-MFN1 (sc-100561, Santa Cruz Biotechnology, Inc), anti-MFN2 (sc-100560, Santa Cruz Biotechnology, Inc), anti-DRP1 (sc-101270, Santa Cruz Biotechnology, Inc), and anti-FIS1 (GTX111010, GeneTex, Inc) antibodies, followed by incubation with secondary antibodies at an appropriate dilution. The blots were visualized with the Western Lightning Plus-ECL (PerkinElmer, Waltham, MA, USA).

### 2.4. Mitochondrial Fragmentation Index

Mitochondrial morphology was assayed as previously described with some modifications [14,18]. Briefly, mitochondria were stained by the fluorescent dye Mitotracker red (Invitrogen, Life Technologies, Carlsbad, CA, USA) and observed under a fluorescence microscope (Olympus BX-UCDB-2; Olympus Corporation, Tokyo, Japan). Mitochondrial morphology was scored in a blinded and independent manner by two experienced technicians. Mitochondria with punctated or fragmented patterns were termed as the fission type. The fission rate (%) was defined as the percentage of fission type mitochondrion among all counted mitochondria, i.e., fission events/total (fission + intermediate + fusion types) × 100. 

### 2.5. Cellular Apoptotic Assay

In the terminal deoxynucleotidyl transferase dUTP nick end labelling (TUNEL) assay, HK-2 cells incubated for 24 h in the presence or absence of empagliflozin were fixed in 4% paraformaldehyde and treated with a blocking solution (10% FBS in PBS) for 30 min at 25 °C. Cells were permeabilized with 0.1% (*v*/*v*) Triton X-100 in PBS, followed by incubation with the TUNEL reaction mixture (Roche, Mannheim, Germany) for 1 h at 37 °C. After the treatment, the cells were observed under a fluorescence microscope (Olympus BX-51; Olympus Corporation, Tokyo, Japan).

### 2.6. Measurement of Mitochondrial Membrane Potential (MMP)

Rhodamine 123 (Invitrogen, Life Technologies, Carlsbad, CA, USA) was used to measure MMP, as previously described [19]. Briefly, 2 × 10^5^ cells were plated in each well of a six-well plate and allowed to attach for 16–18 h. After treatment for 48 h, the cells were harvested with trypsin, washed with PBS, and resuspended in 200 ng/mL of Rhodamine 123. After incubation for 30 min at 37 °C, the cells were washed thrice and resuspended in 500 mL of PBS. Cytofluorimetric analysis was performed using a BD Biosciences FACScan system.

### 2.7. Quantification of Cellular and Mitochondrial ROS

Cellular ROS level was detected using 2′,7′-dichlorodihydrofluorescein diacetate (DCFH-DA) as previously described [20]. Briefly, 2 × 10^5^ cells were incubated with appropriate concentrations of glucose and empagliflozin for 24 h. The cells were washed twice with PBS and incubated with 10 μM DCFH-DA (Molecular Probes, Life Technologies, Carlsbad, CA, USA) for 30 min at 37 °C. Cells were harvested with trypsin/ethylenediaminetetraacetic acid (EDTA), centrifuged at 200× *g* for 5 min, and resuspended in 0.5 mL of PBS. Cellular ROS level was measured by flow cytometry using a fluorescence-activated cell scanner FACScan machine (BD Biosciences Inc., San Jose, CA, USA). The mean fluorescence intensity (MFI) for each sample was calculated using the CellQuest software. Mitochondrial ROS level was calculated using MitoSOX Red (Invitrogen, Life Technologies, Carlsbad, CA, USA). Briefly, 2 × 10^5^ cells were plated in each well of six-well plates and allowed to attach for 16–18 h. After being treated with drugs for 24 h, the cells were harvested by trypsin treatment, washed in PBS, and resuspended in 5 μM MitoSOX red. After incubation for 30 min at 37 °C, the cells were washed thrice and resuspended in 500 μL of PBS. Cytofluorimetric analysis data were similar to those of cellular ROS quantitation.

### 2.8. Measurement of ATP Generation

ATP content was colorimetrically determined with the ATP Colorimetric/Fluorometric Assay kit (BioVision, Milpitas, CA, USA) following the manufacturer’s instructions. Briefly, 1 × 10^6^ cells were maintained in a T75 flask overnight. Treatment groups included 5 mM glucose, 30 mM glucose, 30 mM glucose + 100 nM empagliflozin, and 30 mM glucose + 500 nM empagliflozin groups. After overnight treatment, the cells were harvested, collected by centrifugation at 200× *g* for 3 min, washed with PBS, and lysed in 100 μL ATP assay buffer. The samples or standards (50 μL per well) and the reaction mix (50 μL per well) were added to a 96-well plate and incubated for 30 min. Then, absorbance was measured at 570 nm using an ELISA reader Quant (BioTek Instruments Inc., Winooski, VT, USA)

### 2.9. Statistical Analysis

All experiments were repeated at least three times. The quantitative data are presented as mean ± standard error of the mean (SEM). Statistical analyses were performed using IBM SPSS Statistics for Windows, Version 19.0. (IBM Corp., Armonk, New York, NY, USA). Comparisons between multiple groups were carried out using one-way analysis of variance (ANOVA). Pair-wise comparisons were performed using the *t*-test. A value of *P* < 0.05 was considered significant.

## 3. Results

### 3.1. Empagliflozin Exerts Insignificant Effects on Cell Viability

The viability of HK-2 cells treated with the indicated concentrations of glucose and empagliflozin is shown in Figure 1. We tested empagliflozin at concentrations of 100, 500, 1000, and 2000 nM under normal (5 mM) and high glucose (30 mM) conditions. Treatment with the fixed concentration of acetonitrile, which was used as the solvent for empagliflozin, was considered as vehicle control. Empagliflozin did not affect cellular viability. In comparison with the normal glucose condition, high glucose condition caused a 12% reduction in cell viability (Figure 1). In addition to demonstrating that empagliflozin exerts insignificant effects on cell viability, we performed validation experiments including glucose uptake and RNA interference to silence *SGLT2*. These validation experiments were to confirm the primary effects of empagliflozin on HK-2 cells and to support that our pharmacology experiment results were specifically due to SGLT2 inhibition (Appendix A).

### 3.2. Empagliflozin Rescues High Glucose-Induced Mitochondrial Fragmentation in Human PTCs

Earlier, we have demonstrated the effect of high glucose on mitochondrial fragmentation in human PTCs [14]. We, therefore, investigated whether empagliflozin rescues this effect. Figure 2A–D shows the morphological features of mitochondria from the four treatment groups. Magnified images of indexed mitochondria in each treatment group are shown in Figure 2E–H. We found that HK-2 cells were susceptible to high glucose-induced mitochondrial fragmentation, and empagliflozin could rescue mitochondrial fragmentation. To quantitatively analyze the process of mitochondrial fragmentation, we calculated the mitochondrial fission rate in each treatment group. High glucose (30 mM) condition led to a four-times higher fission rate in HK-2 cells than the normal glucose condition. In the presence of high glucose, empagliflozin at either 100 or 500 nM concentration could reduce the mitochondrial fission rate to a level comparable to that reported under normal glucose condition (Figure 2I).

### 3.3. Impacts of High Glucose and Empagliflozin on the Expression of Mitochondrial Fusion/Fission Proteins

Western blot analysis (Figure 3A) revealed the high glucose-induced downregulation in the expression of fusion proteins (i.e., MFN1 and MFN2, Figure 3B,C). Empagliflozin could reverse the high glucose-induced downregulation of MFN1 and MFN2 expression at 500 and 100 nM concentration, respectively (Figure 3B,C). High glucose failed to cause any changes in the expression of fission proteins (i.e., DRP1 and FIS1, Figure 3D,E). In the presence of high glucose, empagliflozin could downregulate the expression of DRP1 at 100 nM concentration. Similar effects were observed on FIS1 expression in the presence of 500 nM empagliflozin (Figure 3D,E). As mitochondrial morphology is controlled by the balance between fusion/fission proteins, the results shown in Figure 3 support the observation illustrated in Figure 2.

### 3.4. Empagliflozin Reduces the High Glucose-Induced Cellular Apoptosis of HK-2 Cells

Compared to the normal glucose condition (Figure 4A), high glucose condition induced a marked increase in the level of cellular apoptosis (Figure 4B). Empagliflozin at 100 and 500 nM concentrations offered effective protection against high glucose-induced apoptosis (Figure 4C and 4D). Quantitative analysis of TUNEL assay result showed that high glucose markedly induced apoptosis and that empagliflozin could ameliorate this effect (Figure 4E).

### 3.5. Empagliflozin Improves Mitochondrial Function in High Glucose-Treated HK-2 Cells

We examined mitochondrial function of HK-2 cells by analyzing cellular (Figure 5A) and mitochondrial ROS production (Figure 5B), MMP (Figure 5C), and ATP generation (Figure 5D). High glucose increased both cellular and mitochondrial ROS levels. Empagliflozin at 500 nM could reduce the cellular ROS level, although 100 nM empagliflozin was not sufficiently effective (Figure 5A). Empagliflozin at 100 and 500 nM concentrations could maintain the mitochondrial ROS production level (Figure 5B). High glucose caused a decrease in MMP and this effect was ameliorated by empagliflozin at 500 nM concentration (Figure 5C). Empagliflozin at a concentration of either 100 or 500 nM was able to increase the level of ATP in HK-2 cells (Figure 5D). Taken together, the results of mitochondrial functional assays show that empagliflozin at 500 nM concentration produced healthier mitochondria to combat the stress induced by high glucose.

## 4. Discussion

Complimentary to the traditional ‘glomerulocentric’ perspective of diabetic nephropathy, the understanding of the pathophysiological perturbations in the tubules is essential for the successful management of DKD. In the diabetic milieu, increased energy demands and a decreased perfusion result in tubular hypoxia. This phenomenon may drive the development of tubular atrophy and interstitial fibrosis in a vicious cycle that worsens the clinical course of DKD [21]. At the subcellular level, the mitochondrion of diabetic proximal tubules is the key organelle that modulates its propensity to hypoxic injury and oxidative stress. Studies on subcellular levels often unravel the underlying mechanisms that clinical trials may not reach. Here, we investigated the subcellular mechanisms underlying the protective effect of empagliflozin from high glucose-mediated injuries on PTCs. As oxidative stress plays pathologic roles in diabetic kidneys [22,23,24] and empagliflozin may reduce this effect [25,26], we analyzed the direct effects of empagliflozin on the mitochondria of PTCs.

In PTCs, high glucose treatment induces mitochondrial fragmentation and cellular apoptosis [14,27]. Our results showed that empagliflozin protected PTCs from high glucose-induced injuries through the modulation of the expression of mitochondrial fusion and fission proteins. We demonstrate that empagliflozin could rescue high glucose-induced mitochondrial fragmentation. At the protein level, high glucose condition significantly downregulated the expression levels of both mitochondrial fusion proteins, while empagliflozin treatment rescued this effect by enhancing the expression levels of both mitochondrial fusion proteins. Although not significant, high glucose seemed to cause an upregulation in the expression levels of mitochondrial fission proteins, which were significantly downregulated by empagliflozin. In line with the proposed protective mechanism in a different cellular aspect [28], empagliflozin might make mitochondria more resistant to high glucose-driven fragmentation by upregulating the expression of fusion proteins and downregulating the levels of fission proteins. This protective mechanism may explain why empagliflozin protected PTCs from high glucose-induced mitochondrial fragmentation and cellular apoptosis.

In addition to shaping the mitochondrial morphology, empagliflozin could also improve mitochondrial functions. Mitochondrial dysfunction has been reported in experimental diabetic animal and cellular models [29,30,31]. These disturbances in mitochondrial functions may result in the excessive accumulation of cellular ROS [30,32], ATP depletion [33], and MMP reduction [30]. Our results confirm these observations in response to high glucose-induced mitochondrial dysfunctions in PTCs and demonstrate that empagliflozin could ameliorate these effects. In DKD, ROS is produced from various organelles within the cell, including the mitochondrion, endoplasmic reticulum, and the enzyme systems such as Nox. ROS underproduction may be indicative of the mitochondrial dysfunction. Steady-state levels of ROS are involved in normal cellular signaling, whereas excessive ROS production leads to cellular dysfunction [34]. Our results demonstrate that empagliflozin reduced the cellular ROS level while maintaining the mitochondrial ROS production.

The mitochondrion has been recognized as a crucial target for DKD treatment [35,36]. Several mitochondrion-targeted agents, including MitoQ [37], Ebselen [38], Bendavia [39], and MitoTEMPO [40], have been reported. Although the molecular pathways underlying the empagliflozin-mediated increase in mitochondrial fitness are not completely understood, our results indicated that empagliflozin could serve as a mitochondrion-targeting agent. Proximal tubules, the direct site of action of empagliflozin, emerge as a novel therapeutic target for DKD [21]. This study revealed a novel mechanism explaining the renal benefits of empagliflozin, as demonstrated in the EMPA-REG OUTCOME trial, and this warrants future studies to identify its role in diabetic tubulopathy.

## 5. Conclusions

In conclusion, we demonstrated that empagliflozin treatment improves mitochondrial performance in HK-2 cells, as reflected from their morphologies and functions, to overcome high glucose-induced injuries.

## Figures and Tables

**Figure 1 cells-08-01085-f001:**
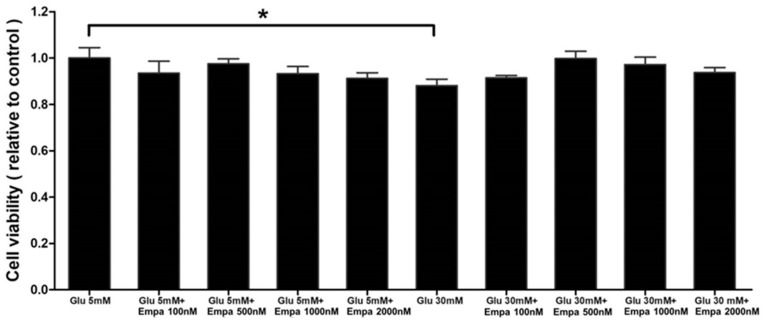
Empagliflozin has no negative effects on the viability of HK-2 cells. Cell viability was measured with crystal violet assay and the absorbance was analyzed at 570 nm using a microplate reader. Data were obtained from three independent experiments and are expressed as mean ± SEM. Only high glucose (30 mM) treatment led to a decrease in cell viability. * *P* < 0.05 versus the normal glucose (5 mM) treatment group. Glu, glucose; Empa, empagliflozin.

**Figure 2 cells-08-01085-f002:**
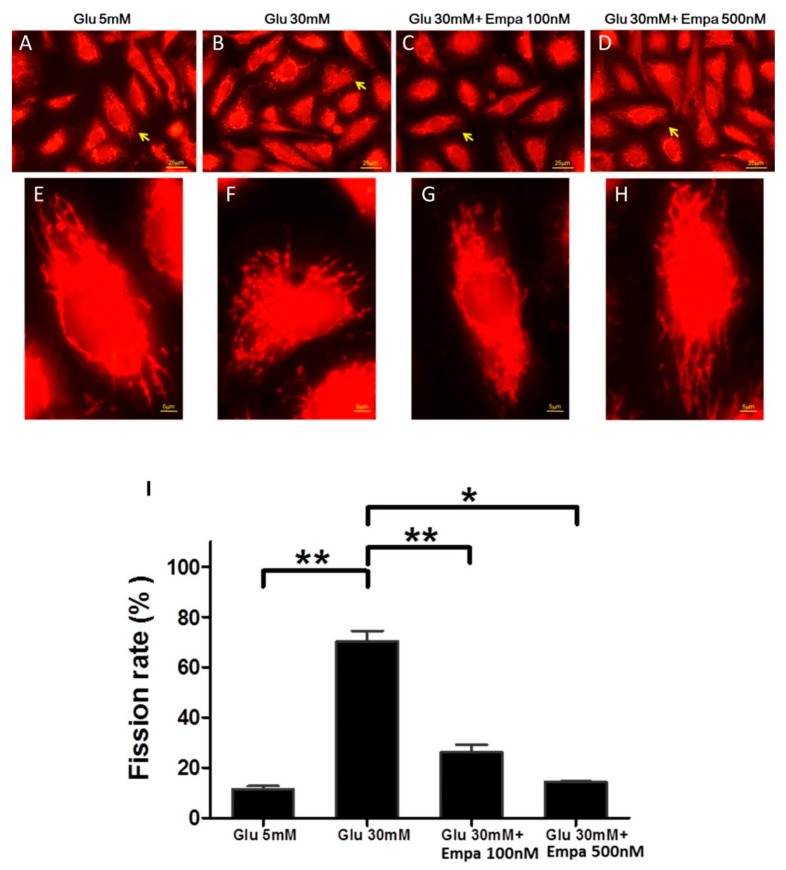
Empagliflozin rescues high glucose-induced mitochondrial fragmentation in human PTCs. HK-2 cells were cultured in 5 mM glucose (**A**), 30 mM glucose (**B**), 30 mM glucose with 100 nM empagliflozin (**C**), and 30 mM glucose with 500 nM empagliflozin (**D**). Mitochondria were stained with Mitotracker red. Magnified images of indexed mitochondria (arrow) in each treatment group are shown in (**E**–**H**). (**I**) The mitochondrial fission rate was higher in high glucose condition. Empagliflozin rescued this effect. Data were obtained from three independent experiments and are expressed as mean ± SEM. * *P* < 0.05, ** *P* < 0.001.

**Figure 3 cells-08-01085-f003:**
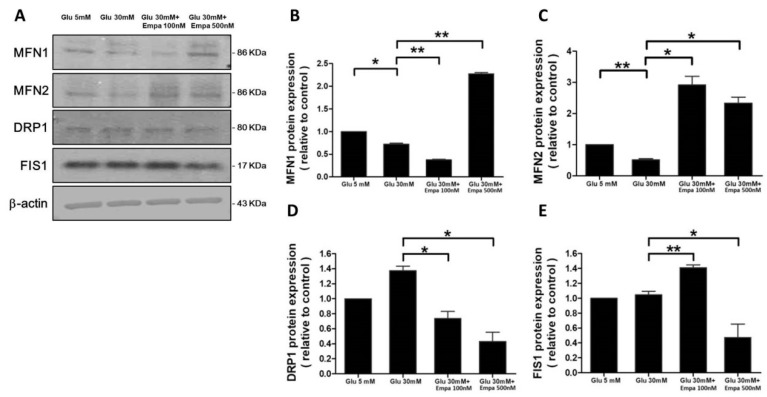
Impact of empagliflozin on high glucose-induced alterations in the expression levels of mitochondrial fusion/fission proteins. (**A**–**E**) The expression of MFN1, MFN2, DRP1, and FIS1 was analyzed with western blotting and normalized to the level of β-actin. Data are expressed as mean ± SEM. * *P* < 0.05, ** *P* < 0.001.

**Figure 4 cells-08-01085-f004:**
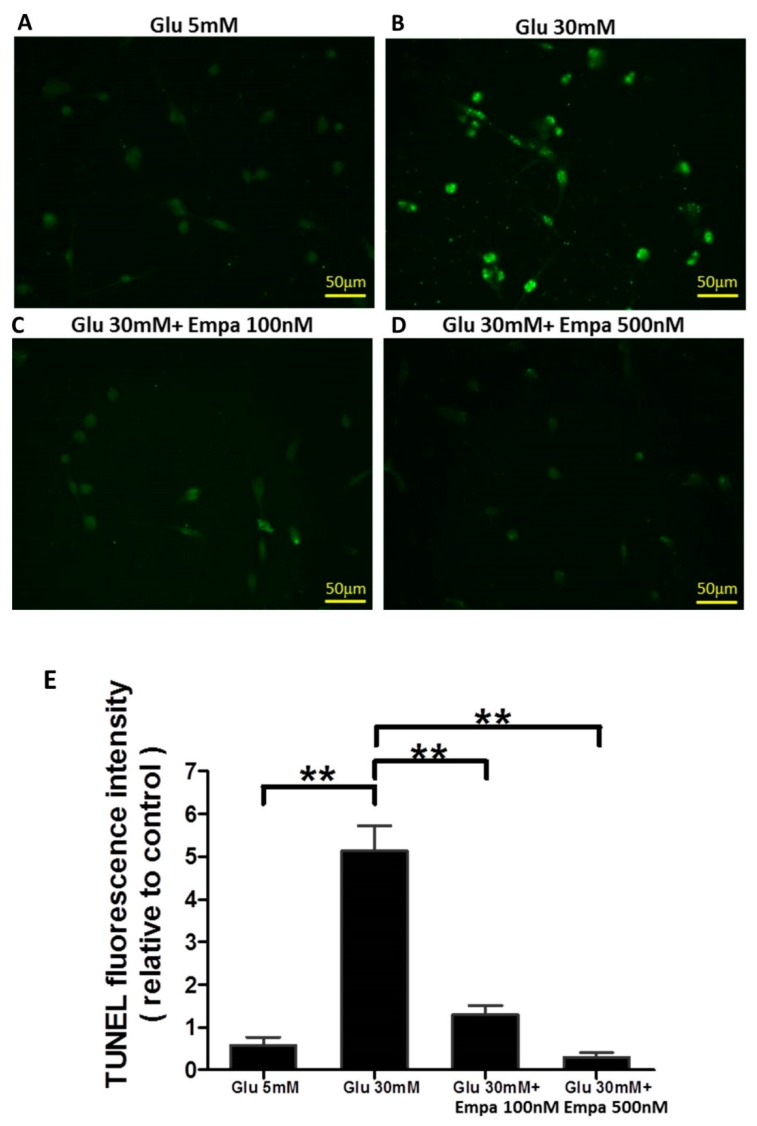
Empagliflozin reduces the high glucose-induced apoptosis of HK-2 cells. (**A**–**D**) Fluorescence images show positive TUNEL staining in the four treatment groups. (**E**) Quantitative analysis of TUNEL assay shows that high glucose markedly induces apoptosis and empagliflozin ameliorates this effect. Data are expressed as mean ± SEM. ** *P* < 0.001.

**Figure 5 cells-08-01085-f005:**
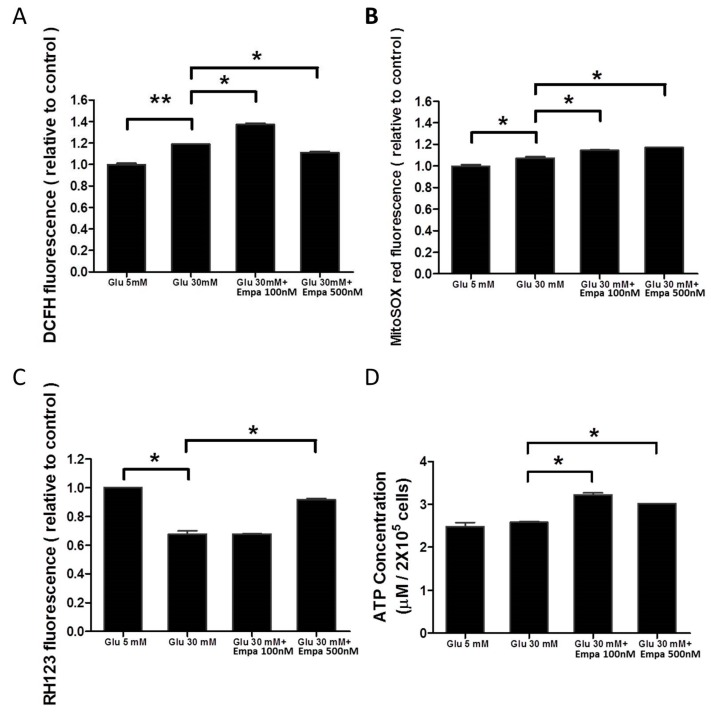
Empagliflozin improves the mitochondrial function of high glucose-treated HK-2 cells. (**A**) Cellular ROS production, (**B**) mitochondrial ROS production, (**C**) MMP, and (**D**) ATP generation in the four treatment groups. Data were obtained from three independent experiments and are expressed as mean ± SEM. * *P* < 0.05, ** *P* < 0.001.

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
