# Peer review of "Empagliflozin Protects HK-2 Cells from High Glucose-Mediated Injuries via a Mitochondrial Mechanism"

_cells, 2019, doi:10.3390/cells8091085_

Round 1

Reviewer 1 Report

In this report, Lee et al. use cellular approaches to examine the protective actions of Empagliflozin (SGLT2 antagonist) on mitochondrial morphology and function.  Overall, a very sound set of experiments conducted in confluent HK-2 (proximal tubule) cells.  The investigators find that cells incubated with high glucose have increased apoptosis and mitochondria fission (with increased fission related proteins).  Mitochondrial and cellular ROS production was measured by fluorescent indicators, as well as, ATP production and membrane potential.   All of these measures pointed to improved mitochondria functionality with empagliflozin (with high glucose).  The mechanism whereby empagliflozin improves mitochondrial well-being is uncertain.  Primarily it is uncertain whether these cells are cultured under conditions that would allow glucose flux.  It is not known what the glucose concentrations were inside the cells.  Does the empagliflozin block transport in these cells?  Do the cells swell under any conditions?  Answering a few of these questions would improve the manuscript significantly. 

1.Methods- Why were homogenates centrifuged at such a high speed (14,000g) with discarding of the pellet? Wouldn’t this speed also discard much of the mitochondria?

2.It is unclear whether cells were cultured on semi-permeable filters to allow for polarization and transport of glucose. Please clarify methods.

3.Intracellular glucose levels should be measured and reported.

4.Expression and/or activity of SGLT2 should be reported.

5.Is anything added to the 5 mM glucose medium to increase the osmolality to the equivalent level as in the 30 mM medium?

6.Does empagliflozin enter the cell?

Reviewer 2 Report

Fitst, authors must declare that this study used a cell line and did in-vitro experiments only in the title and the conclusion. They do not confirm their findings from PTC line in vivo.

Second, SGLT-1 should be mentioned. Does the PTC line have it ? If the answer is yes, authors need to investigate how SGLT-1 work in their experiments. Finally, the authors must show reasons of concentrations of high glucose and empagliflozin. I know 30mM glucose is used for acute glucotoxicity, but I do not know it is adequate for experiments of chronic microvascular complications. If it is used, it is necessary to confirm the findings in-vivo studies. What is the serum concentration of empagliflozine ? The medical package insert showed that 1,000nM 2 hours and around 100nM 24 hours after administration in daily 25 mg users and 500nM 1.5 hours and around around 50nM 24 hours after administration in daily 10 mg users. Why were used 100nM and 500nM mainly.

Round 2

Reviewer 1 Report

The authors have satisfactorily addressed all my concerns.  I suggest acceptance in the present form. 

Reviewer 2 Report

The revised version is better.